# Functional Characterization of Novel Bony Fish Lipoxygenase Isoforms and Their Possible Involvement in Inflammation

**DOI:** 10.3390/ijms232416026

**Published:** 2022-12-16

**Authors:** Sophie Roigas, Dagmar Heydeck, Hartmut Kuhn

**Affiliations:** Department of Biochemistry, Charité-Universitätsmedizin Berlin, Corporate Member of Freie Universität Berlin and Humboldt Universität zu Berlin, Charitéplatz 1, 10117 Berlin, Germany

**Keywords:** eicosanoids, bony fish, enzyme kinetics, polyenoic fatty acids, inflammation

## Abstract

Eicosanoids and related compounds are pleiotropic lipid mediators, which are biosynthesized in mammals via three distinct metabolic pathways (cyclooxygenase pathway, lipoxygenase pathway, epoxygenase pathway). These mediators have been implicated in the pathogenesis of inflammatory diseases and drugs interfering with eicosanoid signaling are currently available as antiphlogistics. Eicosanoid biosynthesis has well been explored in mammals including men, but much less detailed information is currently available on eicosanoid biosynthesis in other vertebrates including bony fish. There are a few reports in the literature describing the expression of arachidonic acid lipoxygenases (ALOX isoforms) in several bony fish species but except for two zebrafish ALOX-isoforms (zfALOX1 and zfALOX2) bony fish eicosanoid biosynthesizing enzymes have not been characterized. To fill this gap and to explore the possible roles of ALOX15 orthologs in bony fish inflammation we cloned and expressed putative ALOX15 orthologs from three different bony fish species (*N. furzeri, P. nyererei, S. formosus*) as recombinant N-terminal his-tag fusion proteins and characterized the corresponding enzymes with respect to their catalytic properties (temperature-dependence, activation energy, pH-dependence, substrate affinity and substrate specificity with different polyenoic fatty acids). Furthermore, we identified the chemical structure of the dominant oxygenation products formed by the recombinant enzymes from different free fatty acids and from more complex lipid substrates. Taken together, our data indicate that functional ALOX isoforms occur in bony fish but that their catalytic properties are different from those of mammalian enzymes. The possible roles of these ALOX-isoforms in bony fish inflammation are discussed.

## 1. Introduction

Eicosanoids are lipid signaling molecules, which have been implicated as pro- and anti-inflammatory mediators in the pathogenesis of inflammation [1,2]. They are biosynthesized via three distinct metabolic pathways: (i) COX pathway, (ii) ALOX pathway, (iii) Cytochrome-p450 pathway. Eicosanoid biosynthesis is well explored in mammals including men [3,4,5] but much less detailed information is currently available on eicosanoid biology in non-mammalian vertebrates. Prostaglandin E4 (PGE4) receptors have been detected in fish [6] and birds [7] and thus PGE4 signaling appears to be important in these non-mammalian vertebrates. From the functional point of view, prostaglandins regulate ovulation in medaka (*Oryzias latipes*) and this data suggested that similar regulatory mechanisms might be involved in ovulation in mammals and bony fish [8]. Even in lower organisms, eicosanoid biosynthesis has been described [9,10,11,12,13].

The lipoxygenase pathway is one of the major eicosanoid-biosynthesizing routes and lipoxygenase isoforms are widely distributed in higher plants [14,15] and animals [16,17]. They are also present in bacteria but here they do not frequently occur. In fact, a systematic search for ALOX genes in bacterial genomes indicated that less than 0.5% of all sequenced bacterial genomes involve putative ALOX genes but no such genes are present in most human pathogenic bacteria. [18]. An important exception is the ALOX-isoform of *P. aeruginosa* [19,20,21,22], which has been suggested as putative pathogenicity factor of the bacterium [20,23].

Vertebrates comprise all animal taxa of the subphylum *Vertebrata*, which include all mammals, fish, birds, reptiles and amphibia. According to recent estimates of the International Union for Conservation of Nature (IUCN) some 74,000 vertebrate species are currently living on earth and almost half of them are fish (Appendix A). Compared with invertebrates (some 1.5 million different species) vertebrates only contribute a minor share (5%) to the extant animal species but bony fish clearly dominate vertebrates [24]. Although ALOX-isoforms have been identified in different bony fish species such enzymes are not widely distributed in these non-mammalian vertebrates. In zebrafish (*D. rerio*), which is frequently employed as model organism in developmental studies [25,26], different ALOX genes have been identified. Two ALOX-isoforms have been expressed as recombinant enzymes and have been characterized with respect to their enzymatic properties [27,28,29]. ALOX activities have also been described in other bony fish species, but little is known about the catalytic properties of the corresponding enzymes. For instance, an ALOX, which oxygenates arachidonic acid to 12-HETE, was detected in gill of the rainbow trout [30]. Later on, the formation of specific ALOX products was detected in different tissues of this bony fish [31]. ALOX isoforms were also described in leukocytes [32] and blood platelets [33] of different trout species. Peripheral leukocytes of different bony fish species produced specific ALOX products when stimulated in vitro with calcium ionophore [34] and similar results were obtained for peripheral blood cells of the lesser spotted dogfish [35]. In 2016 an ALOX5 ortholog was cloned from the large yellow croaker (*L. crocea*) and expression regulation of this enzyme in response of alimentary supplementation with polyunsaturated fatty acids was studied [36]. However, neither of these ALOX-isoforms was characterized as true ALOX15 ortholog.

The human genome involves 6 different ALOX genes (*ALOX15*, *ALOX15B*, *ALOX12*, *ALOX12B*, *ALOXE3*, *ALOX5*) and each of them encodes for a functionally distinct ALOX-isoform [18]. Among the human ALOX-isoforms ALOX15 is somewhat special since it is capable of oxygenating esterified polyenoic fatty acids present in the phospholipids of biomembranes [37,38]. Because of this functional peculiarity we focused our study on bony fish ALOX15 orthologs but did not consider other bony fish ALOX-isoforms. To explore whether real ALOX15 orthologs occur in bony fish we first searched the NCBI protein database for complete ALOX15 sequences and selected three of them (*N. furzeri, P. nyererei, S. formosus*) for recombinant expression and detailed functional characterization. Comparison of the catalytic properties of these enzymes with those of mammalian ALOX15 orthologs indicated similarities but also interesting functional differences. The possible role of these novel enzymes in the pathogenesis of bony fish inflammation will be discussed.

## 2. Results

### 2.1. Recombinant Expression of Putative Bony Fish ALOX15-Isoforms

When we searched the NCBI protein database (https://www.ncbi.nlm.nih.gov/guide/proteins, accessed on 25 April 2019) we found complete cDNA sequences for 7 different putative ALOX15 orthologs and selected the enzymes of *Nothobranchius furzeri* (XP_015813570.1), *Pundamilia nyererei* (XP_005753048.1) and *Scleropages formosus* (XP_018588735.1) for functional characterization. To determine the catalytic properties of the selected bony fish ALOX15-isoforms we first attempted to express the corresponding enzymes as N-terminal his-tag fusion proteins in *E. coli*. Analyzing the bacterial lysate supernatants of *E. coli* cells, transformed with the recombinant expression plasmids of the three bony fish ALOX-isoforms, we found that only the enzyme of *S. formosus* was well expressed in this procaryotic expression system (Figure 1A,D). The ALOX-isoforms of *N. furzeri* and *P. nyererei* were not well expressed as soluble proteins in *E. coli* and thus, we decided to express these enzymes in the baculovirus insect cell system. Here, we found abundant expression of the two ALOX-isoforms (Figure 1B,C) and the purified proteins were immunoreactive (Figure 1D).

When we carried out in vitro activity assays using the cellular lysate supernatant of the *E. coli* cells transformed with the recombinant expression plasmid carrying the coding sequence for the ALOX-isoform of *S. formosus* we observed the formation of conjugated dienes co-migrating in RP-HPLC with an authentic standard of 12-HETE (Figure 2A). When the lysis supernatant was heated for 2 min at 90 °C (heat inactivation control) these products were not formed (Figure 2B).

Similar in vitro activity assays using the cellular lysis supernatants of Sf9 cells infected with the recombinant baculovirus carrying the coding sequences of the ALOX-isoforms of *N. furzeri* and *P. nyererei* indicated the formation of similar oxygenation products (Figure 2D,G). Here, again, product formation was completely abolished when the cellular lysate supernatant was heated for 2 min (Figure 2E,H). For these enzymes we also carried out no-enzyme control incubations, in which PBS was added to the activity assay instead of the cellular lysate supernatant. In these assays we did not observe the formation of conjugated dienes (Figure 2C,F). When we examined the 12-HETE peak formed by the three putative ALOX15 orthologs in more detail we noticed that the shape of these peaks was not perfectly symmetric. In fact, zooming into the product peaks we detected a subtle front-shoulder and this observation suggested that the peaks were not homogenous, but consisted of two different conjugated dienes which are only partly resolved under our chromatographic conditions. To solve this problem we prapared the conjugated dienes formed by the three putative bony fish ALOX15 orthologs and analyzed them by NP-HPLC (Figure 3A,C). With this method the RP-HPLC peak was split into separate peaks of conjugated dienes. 12-HETE was the major arachidonic acid oxygenation product but 8-HETE could be identified as minor side product.

Unfortunately, when corresponding in vitro ALOX activity assays were carried out with the elution fractions of the affinity chromatography (e1–e5) of the three enzymes we did not detect the formation of specific ALOX products. These data suggested that the catalytic activity of the three recombinant enzyme species was lost during the purification procedure although enzyme purification was performed at 4 °C. To explore whether other ALOX-isoforms were also inactivated during the purification procedure we performed similar experiments with human ALOX15 as positive control. Here, we recovered catalytically active enzyme after the purification procedure. Although we attempted to optimize the purification strategy and although we applied alternative purification strategies we did not succeed. The molecular basis for the loss in catalytic activity during affinity chromatography of bony fish ALOX15 orthologs remains unclear but because of this observation we used the cellular lysate supernatants as enzyme source for our characterization experiments.

### 2.2. Product Specificity with Different PUFAs

To explore whether the putative bony fish ALOX15 orthologs also accept other polyenoic fatty acids as substrate we incubated the cellular lysis supernatants with eicosapentaenoic acid (EPA) and docosahexaenoic acid (DHA), prepared the conjugated dienes by RP-HPLC and analyzed the reaction products by NP-HPLC. For comparison identical incubations were carried out with arachidonic acid (AA). From Figure 3A–C it can be seen that AA is oxygenated by the three bony fish ALOX-isoforms to a mixture of 12-HETE and 8-HETE. For the ALOX-isoform from *N. furzeri* 12-HETE was the major AA oxygenation product, which contributed about 85 % to the conjugated dienes formed during the incubation period (Figure 3A). For the enzymes of *P. nyererei* and *S. formosus* higher amounts of 8-HETE were observed (Figure 3B,C) but still 12-HETE formation was dominant.

When EPA was used as substrate (Figure 3D–F) conjugated dienes co-eluting in NP-HPLC with an authentic standard of 12-HEPE were detected as major oxygenation products. GC-MS analysis of the trimethylsilyl derivatives showed major fragmentation ions at *m*/*z* 211 and 324 (alpha cleavage ions) confirming the chemical identity of this product as 12-HEPE. Smaller amounts of an additional conjugated diene, which co-chromatographed in NP-HPLC with an authentic standard of 8-HEPE, were also detected (Figure 3D–F) and this data indicated that the bony fish enzymes also exhibit a dual reaction specificity when oxygenating EPA.

DHA is not a good substrate for the ALOX-isoforms from bony fish and the enzyme of *P. nyererei* only formed about 10% of the conjugated dienes that were synthesized from EPA. However, the formation of small amounts of conjugated dienes was observed and when we analyzed the major oxygenation products formed from this substrate, 14-HDHA was expected as major oxygenation product. NP-HPLC analysis of the reaction products confirmed this conclusion (Figure 3G,H). Surprisingly, similar amounts of 10-HDHA were formed and this data indicates a pronounced dual reaction specificity of these enzymes with the highly unsaturated DHA.

For control activity assays a lysate supernatant of *E. coli* cells was used, which were transformed with the empty expression plasmid. A similar control was carried out using the cellular supernatant of Sf9 cells infected with a control baculovirus and the outcomes of these negative controls are shown in Appendix A (supplement).

### 2.3. Temperature Dependence and Determination of the Activation Energy

The rate of a chemical reaction is usually increased at elevated temperatures. However, for enzymatic reactions the temperature-dependent increase in reaction rate is counteracted by heat-induced protein denaturation at higher temperatures. Thus, most enzymes exhibit an optimal temperature (T_opt_) for their catalytic reactions. For homeothermic organisms this T_opt_ is usually close to internal temperatures of the organism. In contrast, for poikilothermic organisms like fish and reptiles, which quickly adapt their body temperature to the temperature of their environment, lower T_opt_ values are meaningful. On the other hand, for microorganisms living in the surrounding of hotsprings enzymes with higher T_opt_ values are required [39,40]. When we determined the T_opt_ of the ALOX-isoforms of the three poikilothermic bony fish species we noticed that for the enzymes of *N. furzeri* and *S. formosus* there were only minor differences in the reaction rate in the temperature interval between 20–35° C (Figure 4A,C). Even at 15° C the enzymes exhibited more than 60% of their optimal catalytic activity. However, at temperatures higher then 35 °C a strong decline of the catalytic activity was observed, which is most probably related to partial enzyme denaturation. For poikilothermic organisms like bony fish species such flat temperature profiles are physiologically meaningful since the catalytic activities of their enzymes should not substancially vary when the organisms adapt to the dfifferent temperatures of their habitates. For the ALOX-isoform of *P. nyererei* (Figure 4B) the temperature profile is somewhat different. Here, a temparature dependent decline of the catalytic activity was already observed at 30 °C. This data suggests that the *P. nyererei* ALOX exhibits a lower thermostability but the molecular basis for this higher heat lability has not been explored.

From the temperature profiles shown in Figure 4 it can be concluded that in the temperature range between 5 °C and 20 °C temperature dependent denaturation may be minimal. Thus, we used these activity data to calculate the activation energies for AA oxygenation. We found (Figure 5) that in this temperature range the activity data followed the Arrhenius equation with correlation coefficients R^2^ > 0.9 (R^2^ for *N. furzeri* = 0.9859, R^2^ for *P. nyererei* = 0.9451, R^2^ for *S. formosus* = 0.9080) and the following numeric values of the activation energies were calculated: 57.3 kJ/mole for *N. furzeri*, 33.2 kJ/mole for *P. nyererei* and 42.5 kJ/mole for *S. formosus*.

These values are higher than the activation energy determined for the soybean LOX-1 [41] and for the quasi-LOX activity of hemoglobin [42]. For the ALOX isoform of *P. aeruginosa* an activation energy of 33.8 kJ/mole has previously been determined [43] and this value is similar to the corresponding value of the *P. nyereri* enzyme (Figure 5B). Moreover, the ALOX15 ortholog of *G. gorilla* [44] exhibits a similar activation energy as the *N. furzeri* enzyme.

### 2.4. pH-Dependence of the Catalytic Activity of Bony Fish ALOX Isoforms

In general, the pH of the reaction buffer impacts the catalytic activities of enzymes since alterations in the pH modify the degree of dissociation of amino acid side chains. These changes are particularly relevant for the imidazole group of His residues since the pK_a_-value of this functional group is close to the physiological pH of the blood plasma (pH = 7.4). The intracellular pH is somewhat lower depending on the functional state of the cell but still it remains in the near neutral range [45]. However, in different subcellular organelles such as lysosomes, phagolysosomes or other intracellular vesicles lower pH values have been determined [46].

From Figure 6 it can be seen that the three bony fish ALOX-isoforms exhibit a similar pH-dependence. For the enzymes of *N. furzeri* (Figure 6A) and *S. formosus* (Figure 6C) pH_opt_-values close to the plasma pH were determined. For the *P. nyererei* enzyme the pH_opt_ was somewhat lower (Figure 6B). Similarly neutral pH_opt_ values were previously determined for rabbit [47] and human ALOX15 [38]. In contrast, soybean LOX-1 shows a pH_opt_ in the more alkaline range [48].

### 2.5. Oxygen Affinity

ALOX-isoforms are oxygen metabolizing enzymes but for the time being no specific oxygen binding sites have been identified for any ALOX-isoform [49]. Most probably, the enzyme bound fatty acid radical, which is formed at the active site of the enzymes during initial hydrogen abstraction, may function as immediate oxygen acceptor. To reach this carbon centered radical, atmospheric dioxygen must travel from the reaction buffer into the catalytic center and oxygen access channels have been suggested in the crystal structures of different ALOX-isoforms [50,51,52,53,54,55]. Functional studies on different mammalian ALOX-isoforms suggested that these enzymes exhibit a high oxygen affinity [56]. On the other hand, a high oxygen K_M_ was determined for *P. aeruginosa* ALOX [21].

To judge the oxygen affinity of the bony fish ALOX-isoforms we carried out comparative activity assays under normoxic (air saturation) and hyperoxic (oxygen saturation) conditions. If the enzymes exhibit a high oxygen affinity, similar AA oxygenase activities should be measured under these conditions. In contrast, if the enzymes exhibit low oxygen affinities a much higher product formation is expected under hyperoxic conditions. As shown in Figure 7A the putative ALOX15 of *N. furzeri* exhibited a higher catalytic activity under hyperoxic conditions. Although the difference was statistically significant (*p* < 0.001) its extent (<50%) was not particularly impressive. For the putative *P. nyererei* ALOX15 we observed a significant drop in the catalytic activity under hyperoxic conditions (*p* < 0.001). A reduced catalytic activity was also found for the enzyme of *S. formosus*. From these data it was not possible to conclude the affinity of the putative bony fish ALOX15 orthologs for oxygen since two of these enzymes appeared to be inactivated under hyperoxic conditions. In general, ALOX-isoforms are oxygen sensitive enzymes, which loose catalytic activity when stored at room temperature. Under anaerobic conditions they are more stable suggesting that oxidation of redox-sensitive amino acids might contribute to the instability.

### 2.6. Oxygenation of Complex Substrates by the Putative Bony Fish ALOX-Isoforms

Oxygenated polyenoic fatty acids such as HETE- and HEPE isomers carry bisallylic methylenes and thus, they constitute ALOX substrates.

To further characterize the substrate specificity of the bony fish ALOX-isoforms we incubated the recombinant enzymes with 5-HETE and 15-HETE and analyzed the major reaction products. We found that 5-HETE was oxygenated by the ALOX-isoforms of *N. furzeri* and *P. nyererei* mainly to 5,12-diHETE (Figure 8A,B). Since this substrate involves a bisallylic methylene at C_10_ of the fatty acid backbone and since the OH-group is far distant from the methyl end of the fatty acid, formation of 5,12-diHETE is plausible. However, in 15-HETE, which also carries a bisallylic methylene at C_10_ the OH-group is close to the methyl end of the fatty acid and this hydrophilic group may favor inverse head-to-tail substrate alignment. If this hypothesis is correct, the formation of 8,15-diHETE should be dominant. Analyzing the major oxygenation products formed from 15-HETE by the ALOX-isoforms of *N. furzeri* and *P. nyererei* we indeed observed dominant formation of 8,15-diHETE (Figure 8C,D).

In addition, we specifically searched for the formation of 14,15-diHETE, which should have been formed when the substrate was tail-first aligned at the active site. However, we only detected small amounts of this compound in RP-HPLC. Interestingly, 5*R/S*-HETE and 15*R/S*-HETE are more efficiently oxygenated by the bony fish ALOX-isoforms than AA (Table 1) and this observation may be related to the better water solubility of the HETE derivative when compared with AA.

For other ALOX isoforms HETE isomers may also function as substrates but here the oxygenation rate was shown to be lower than that of AA oxygenation [57,58]. We also tested 18-HEPE, 5,15-diHETE and 5,6-diHETE as substrates for the bony fish ALOX isoforms but for these eicosanoids we only detected minor substrate conversion (Table 1).

## 3. Discussion

### 3.1. Degree of Novelty, Advancement of Knowledge and Limitations

Eicosanoids are important lipid signaling molecules, which have been implicated in a large number of physiological and patho-physiological processes [3], and ALOX-isoforms contribute to eicosanoid biosynthesis [16,17]. In mammals, eicosanoid biosynthesis has extensively been studied but little is known about eicosanoid biosynthesizing enzymes in other vertebrates, such as birds and bony fish. Although arachidonic acid lipoxygenases have previously been described in zebrafish [27,28,29] and although a few additional literature reports suggested that such enzymes also occur in different cells of other bony fish species [30,31,33,35,36,59] the enzymes themselves have not been characterized. Here, we expressed novel ALOX-isoforms from three different bony fish species as recombinant proteins and characterized them with respect to their catalytic properties. Using polyenoic fatty acids as substrates, these bony fish ALOX-orthologs exhibit dual reaction specificity oxygenating AA to a mixture of 12-HETE and 8-HETE (Figure 3). EPA and DHA are also oxygenated with dual reaction specificity and from these polyenoic fatty acids mixtures of 12- and 8-HEPE (EPA) and 14- and 10-HDHA were analyzed (Figure 3). Mechanistically, this type dual reaction specificity is very different from the dual specificity of mammalian ALOX15 orthologs [60,61] since it involved hydrogen abstraction from a single bisallylic methylene (C_10_ of AA and EPA, C_12_ of DHA) as well as simultaneous [+2] and [−2] rearrangement of the fatty acid radical formed during initial hydrogen abstraction. The dual reaction specificity of mammalian ALOX15 orthologs (simultaneous formation of 12- and 15-HETE) can be explained by the Triad Concept [61,62,63]. According to this mechanistic scenario substrate fatty acids slide into the substrate binding pocket with their methyl end ahead and two bisallylic methylenes (e.g., C_13_ and C_10_ of AA) are localized in close proximity to the enzyme-bound non-heme iron. Thus, hydrogen can be abstracted from two different bisallylic methylenes and two different products (15-HETE and 12-HETE) can be formed. In contrast, the dual reaction specificity of the bony fish ALOX isoforms described here cannot be explained by the Triad Concept since for the formation of 12- and 8-HETE hydrogen is only abstracted from C_10_. Although the mechanistic basis for this enzyme property has not been explored in this study the most probable explanation for the simultaneous formation of 12-HETE and 8-HETE is that the substrate fatty acid is oppositely aligned at the active site. The formation of both products (12HETE and 8-HETE) involves hydrogen abstraction from C_10_ but opposite directions of radical rearrangement ([+2] rearrangement for 12-HETE, [−2] rearrangement for 8-HETE). The mechanistic basis explaining the two types of dual reaction specificities of mammalian ALOX15 orthologs and the bony fish enzymes characterized here are depicted in Figure 9. If this concept is correct the stereochemistry of the two chiral centers of the two products (C_12_ in 12-HETE and C_8_ in 8-HETE) should be opposite. In fact, 12*S*-HETE and 8*R*-HETE should be formed. For this study, we did not perform chiral analyses of the reaction products but this working hypothesis will be tested in follow up experiments.

Our inability to purify the recombinant bony fish ALOX15 orthologs is certainly a limitation of the present study. To overcome this problem, we expressed the enzymes in two different recombinant overexpression systems (*E. coli*, Sf9 cells), attempted to optimize the affinity purification strategy and employed conventional purification methods such as gel filtration. Unfortunately, neither of these attempts were successful and thus, we decided to use crude enzyme preparations for the characterization studies. In principle, the foreign proteins present in the cellular lysate supernatants might impact functional enzyme characteristics but previous experiments have shown that the reaction specificity of ALOX15 [64], ALOX5 [65] and ALOX12 [66] did not depend on the degree of purity of the enzyme preparation. Thus, the purified bony fish ALOX15 orthologs are likely to exhibit a similar reaction specificity as the crude enzyme preparations.

### 3.2. Catalytic Properties of Bony Fish ALOX-Isoforms

Although previous experimental data suggested the existence of ALOX-isoforms in bony fish [27,31,33,63,67] little is known about the catalytical properties of the corresponding enzymes. Two lipoxygenase isoforms of the zebrafish (zfLOX1 and zfALOX2) have previously been expressed as recombinant proteins and have been functionally characterized [27,29]. In this study we expressed putative ALOX15 orthologs of three additional bony fish species as N-terminal his-tag fusion proteins and characterized the recombinant enzymes with respect to their catalytic properties. We found that the enzymes oxygenate AA, EPA and DHA with dual reaction specificity (Figure 3). Monohydroxylated fatty acids such as 5(R/S)-HETE and 15-(R/S)-HETE were also effectively oxygenated, but 18-HEPE and dihydroxy eicosanoids were not good substrates (Table 1).

Bony fish are poikilothermic organisms adapting their body temperature to the temperature of their environment. Thus, the ALOX-isoforms should exhibit a rather broad temperature profile and the reaction rates of the enzymes should not dramatically differ in the temperature range between 15 °C and 30 °C. When we recorded the temperature profiles of AA oxygenation by the different bony fish ALOX orthologs (Figure 4), we did not observe pronounced differences of the catalytic activities in this temperature range. Even at 10 °C the enzymes of *P. nyererei* and *S. formosus* exhibited more than 50% residual activity. This activity deficit can easily be compensated by upregulation of enzyme expression in case habitat temperatures are dropping. From the activity data measured in the temperature range between 5 °C and 20 °C we determined the activation energies for AA oxygenation by the recombinant bony fish ALOX-isoforms (Figure 5) and compared these data with corresponding values of other ALOX-isoforms [41,43,44]. The activation energy of an enzymatic reaction is defined as energy that is required to reach the catalytic transition state. Unfortunately, the transition state of the ALOX reaction has not been structurally defined and thus, the numeric values of the activation energies do not provide deeper insights into the mechansism of fatty acid oxygenation. However, from the mechanism of the ALOX reaction it is most likely that the formation of the transition state involves stereoselective hydrogen abstraction from the bisallylic methylene. Since this is the rate limiting step of the overall ALOX reaction, a part of the activation energy is likely to be used for this elementary reaction of ALOX catalysis.

### 3.3. Evolutionary Aspects of Bony Fish ALOX-Isoforms

*ALOX* genes frequently occur in mammalian genomes and in most mammalian species several functional ALOX are present [61]. The human genome involves 6 different ALOX genes (*ALOX15*, *ALOX15B*, *ALOX12*, *ALOX12B*, *ALOXE3*, *ALOX5*) and each of them encodes for a functionally distinct ALOX-isoform [68,69]. In the murine genome an orthologous gene exists (*Alox15*, *Alox15b*, *Alox12*, *Alox12b*, *Aloxe3*, *Alox5*) for each human ALOX gene but in addition, a functional Aloxe12 gene has also been identified [68,69]. This gene encodes for an epidermal AA 12S-lipoxygenating enzyme [70], which shares a high degree of amino acid identity with mouse Alox15. Except for Aloxe12 systemic and/or conditional knockout mice are currently available and functional studies with these animals indicated that each Alox-isoform has its unique biological function [71,72,73,74,75,76,77]. Expression silencing of one Alox-isoform cannot be compensated by upregulation of other ones. Thus, the structural diversity of mouse Alox-isoforms may not be interpreted as indication for functional redundancy of the mouse genome.

Bony fish (*Osteichthyes*) represent the largest taxonomic superclass of the subphylum *Vertebrata* [24]. With about 32,000 different species, bony fish clearly outnumber birds, reptiles, amphibians and mammals (Appendix A). Unfortunately, little is known about the lipoxygenase pathway of bony fish. The zebrafish genome involves seven different ALOX genes (*zfALOX1, zfALOX2, zfALOX3a, zfALOX3b, zfALOX3c, zfALOX4a, zfALOX4b)* and four of them (*zfALOX2, zfALOX3a, zfALOX3b, zfALOX3c*) share a high degree of sequence conservation with the human and mouse ALOX5 gene [27]. When expressed as N-terminal his-tag fusion protein zfALOX2 converted AA dominantly to 5S-HETE. The finding that functional ALOX5 orthologs are expressed in zebrafish together with the observation that most other human leukotriene relevant genes have an ortholog in the zebrafish genome, suggests the biological relevance of leukotriene signaling in this bony fish. zfALOX1 encodes for an AA 12S-lipoxygenating enzyme but sequence comparisons and functional characterization of the recombinant enzyme did not allow a clear-cut assignment of this zfALOX isoform to any of the mouse and/or human ALOX-isoforms. In fact, this enzyme shares a similar degree of amino acid sequence identity (40–47%) with all human and mouse ALOX-isoforms. Expression silencing of the zfALOX1 gene using the morpholino technology induced developmental defects in the zebrafish brain [28] but such defects have not been described for any mouse Alox-isoform. Here, we report that the putative ALOX15 orthologs exhibit a dual reaction specificity which is quite different from that of mammalian ALOX15 orthologs. Moreover, these enzymes do not follow the Triad Concept explaining the reaction specificity of all mammalian ALOX15 orthologs tested so far [61]. Thus, on the basis of the currently available structural and functional data it is impossible to define whether the ALOX-isoforms characterized in this study represent functional equivalents of the mammalian ALOX15 orthologs. According to our opinion more functional data are required to confirm the database annotation of these enzymes as ALOX15 orthologs.

### 3.4. Possible Roles of ALOX-Isoforms in Bony Fish Inflammation

In mammals ALOX-isoforms have been implicated in the biosynthesis of pro- and anti-inflammatory mediators [1,2]. ALOX5 orthologs, which also occur in bony fish [27,36], are key enzymes in the biosynthesis of pro-inflammatory leukotrienes [16] and the formation of leukotriene B4 by bony fish leukocytes has also been reported [78]. In mammals, arachidonic acid 12- and 15-lipoxygenating ALOX-isoforms have been implicated in the biosynthesis of anti-inflammatory and pro-resolving mediators such as lipoxins, resolvins and maresins [79] and lipoxin formation has also been reported in bony fish [34,35]. In vitro, arachidonic acid 15-lipoxygenating ALOX-isoforms exhibit a higher biosynthetic capacity for pro-resolving lipoxins when compared with arachidonic 12-lipoxygenating enzymes [80] and similar results have recently been presented for the biosynthesis of resolvin E4 [61]. However, to the best of our knowledge arachidonic acid 15-lipoxygenating ALOX-isoforms have not been reported in bony fish and thus, the biosynthetic capacity for these pro-resolving mediators should be compromised. When we searched the bony fish database for ALOX15 orthologs we extracted genomic sequences encoding for putative ALOX15 orthologs but expression and functional characterization of the corresponding enzymes indicated that they do not convert arachidonic acid to 15-HETE. Instead, 12-HETE and 8-HETE are the major arachidonic acid oxygenation products (Figure 3). However, these metabolites cannot be further converted to lipoxins. It may of course be possible, that the bony fish ALOX-isoforms characterized in this study exhibit a different reaction specificity with complex substrates as it has recently been reported for the arachidonic acid 8-lipoxygenating Alox15b of mice [81]. When this enzyme was incubated in vitro with nanodiscs serving as membrane mimetics, the major arachidonic acid oxygenation product was 15-HETE [81]. The bony fish ALOX-isoforms characterized here also exhibit an arachidonic acid 8-lipoxygenating activity (Figure 3) and thus, they might also form 15-HETE when nanodiscs or authentic biomembranes are offered as substrate. Unfortunately, corresponding experiments have not been carried out in the present study but they are underway in our research group. If this hypothesis can be supported by direct experimental data the biosynthetic pathway for pro-resolving lipoxins in bony fish could be explained more easily.

## 4. Materials and Methods

### 4.1. Chemicals

The chemicals used for the different experiments were obtained from the following sources: Phosphate buffered saline without calcium and magnesium (PBS) from PAN Biotech (Aidenbach, Germany); nitrocellulose blotting membrane from Serva Electrophoresis GmbH (Heidelberg, Germany); EDTA (Merck KG, Darmstadt, Germany), arachidonic acid (AA), eicosapentaenoic acid (EPA), docosahexaenoic acid (DHA) and authentic HPLC standards of HETE-isomers (15R/S-HETE, 12S/R-HETE, 8R/S-HETE, 5S-HETE), HEPA-isomers (12R/S-HEPA, 8R/S-HEPA), HDHA-isomers (14R/S-HDHA, 10R/S-HDHA) from Cayman Chem (distributed by Biomol GmbH, Hamburg, Germany); 5R/S-HETE, 15R/S-HETE, 18S-HEPE, 18R-HEPE, 5S,15S-diHETE and 5S,6R-diHETE used as substrate for the bony fish ALOX-isoforms (Table 1) were also purchased from Cayman Chem (distributed by Biomol GmbH, Hamburg, Germany). Acetic acid from Carl Roth GmbH (Karlsruhe, Germany); sodium borohydride from Life Technologies, Inc. (Eggenstein, Germany); isopropyl-β-thiogalactopyranoside (IPTG) from Carl Roth GmbH (Karlsruhe, Germany); restriction enzymes from ThermoFisher (Schwerte, Germany); the *E. coli* strain Rosetta2 DE3 pLysS from Novagen (Merck-Millipore, Darmstadt, Germany). The Bac-to-Bac baculovirus expression system was purchased from Invitrogen Life Technologies (ThermoFisher, Schwerte, Germany). Oligonucleotide synthesis was performed at BioTez Berlin Buch GmbH (Berlin, Germany). Nucleic acid sequencing was carried out at Eurofins MWG Operon (Ebersberg, Germany). HPLC grade methanol, acetonitrile, n-hexane, 2-propanol, ethanol and water were from Fisher Scientific (Waltham, MA, USA).

### 4.2. Database Searches and Amino Acid Alignments

The cDNAs encoding for putative ALOX15 orthologs of *Nothobranchius furzeri* (XP_015813570.1), *Pundamilia nyererei* (XP_005753048.1) and *Scleropages formosus* (XP_018588735.1) were extracted from the NCBI protein database (https://www.ncbi.nlm.nih.gov/guide/proteins, accessed on 25 April 2019). It should be stressed that the original putative ALOX15 sequence of *S. formosus* (XP_018588735.1) was updated in May 2019 and that this enzyme was annotated as ALOX5 (XP_018588735.2).

### 4.3. Cloning of Bony Fish ALOX Isoforms

The coding sequences of the putative ALOX15 sequences of *Nothobranchius furzeri* (XP_015813570.1), *Pundamilia nyererei* (XP_005753048.1), *Scleropages formosus* (XP_018588735.1) were extracted from the NCBI protein database and were chemically synthesized (BioCat GmbH, Heidelberg, Germany). For further subcloning from the initial puC57 synthesis vector into the pET28b+ (Novagen/Merck, Darmstadt, Germany) or pFastBac HT expression vectors, a SalI restriction site was introduced immediately upstream of the internal start codon and a HindIII restriction site was generated immediately downstream of the stop codon. The sequences were optimized for bacterial expression by introduction of silent mutations. Finally, the recombinant plasmids were digested with SalI and HindIII to check for the ALOX insert and for each bony fish species a positive clone involving the ALOX insert was sequenced (Eurofins Genomics Germany GmbH, Ebersberg, Germany).

### 4.4. Expression of Bony Fish ALOX Isoforms

Expression of the enzymes as N-terminal hexa-his-tag fusion proteins was carried out as described in [82]. Briefly, *E. coli* Rosetta2 (DE3)-pLysS cells were transformed with the recombinant pET28b plasmids and bacteria were grown on agar plates containing kanamycin and chloramphenicol as selection markers. An isolated clone was picked and bacteria were grown as liquid cultures in a glucose free MSM with added trace elements and glucoamylase. Protein expression was induced by the addition of IPTG. After 18 h cells were harvested, homogenized by sonication and the cell free lysis supernatant was used as enzyme source.

For those ALOX isoforms, which were not well expressed in *E. coli*, the coding sequences of the cDNAs were subcloned into the pFastBac HT vector and the corresponding bacmids as well as the recombinant baculoviruses were generated according to the manufacturer’s instructions (Bac-to-Bac^®^ Baculovirus Expression System, Invitrogen Life Technologies/Thermo Fisher, Schwerte, Germany). Sf9 cells were infected with different amounts of the recombinant baculoviruses and incubated on a shaking platform until about 30% of the cells had died (on average after 72 h, tested by trypane blue exclusion). Next, the cells were harvested, lysed by sonication and the cell free lysate supernatant was used as enzyme source.

### 4.5. Ni-Agarose Based Affinity Purification of Recombinant ALOX-Isoforms

Recombinant his-tag fusion proteins were attempted to purify by affinity chromatography on a Ni-agarose column. For this purpose, 0.4 mL of Ni-NTA agarose gel (Bio-Rad, Hercules, CA, USA) was transferred into a 5 mL Ependorf tube and was washed three times with 3 mL of PBS. Then, 1 mL of the cellular lysate supernatant was added and the solution was incubated for 60 min at 4 °C under gentle shaking. The gel suspension was transferred into an empty column and the chromatographic flow-through was collected. The gel was rinsed with 0.5 mL of low imidazole (10 mM imidazole) rinsing buffer (0.1 M Tris-HCl buffer, pH 8.0 containing 300 mM NaCl) and subsequently with two consecutive rinsing steps (0.5 mL each) using higher imidazole (25 mM imidazole) rinsing buffer. Finally, the his-tag fusion proteins were eluted with 1 mL of high imidazole elution buffer (0.1 M Tris-HCl buffer, pH 8.0 containing 300 mM NaCl and 200 mM imidazole) and fractions of 0.2 mL were collected. Aliquots of the lysate supernanatnts (I, input), of the flowthrough (D) and of the different elution fractions were applied to SDS-PAGE to follow elution of the recombinant his-tag fusion proteins.

### 4.6. Immunoblotting

To identify the expressed his-tag fusion proteins in the different elution fractions of the Ni-NTA agarose affinity chromatography, aliquots of the elution fractions were analyzed by SDS-PAGE and the separated protein bands were transferred to a nitrocellulose membrane (Thermo Fisher Scientific Inc., Waltham, MA, USA) by a wet blotting method. The blots were reversibly stained with Ponceau red (Sigma-Aldrich, St. Louis, MO, USA) to ensure complete protein transfer. Subsequently the blotting membrane was blocked using the blotting-grade blocker solution (Bio-Rad, Hercules, CA, USA) and was then incubated for 1 h with a 1:5000 diluted anti-his-tag antibody carrying horse redish peroxidase as marker enzyme (Miltenyi Biotec GmbH, Bergisch Gladbach, Deutschland). Immunoreactive bands were visualized using the SERVALight Polaris CL HRP WB Substrate Kit (Serva Electrophoresis GmbH, Heidelberg, Germany). Chemiluminescence was detected using a FUJIFILM Luminescent Image Analyzer LAS-1000plus & Intelligent Dark Box II.

### 4.7. In Vitro Activity Assays and HPLC Analyses of the Reaction Products

Enzyme activity assays and analysis of the reaction products by HPLC were carried out as described in [82]. For this purpose, different volumes of the cell-free supernatants were incubated with 100 µM arachidonic acid and the reduced reaction products were analyzed by RP-HPLC on a Shimadzu instrument connected with a Hewlett Packard diode array detector 1040 A. Metabolites were separated on a Nucleodur C_18_ Gravity column (Macherey-Nagel, Düren, Germany; 250 × 4 mm, 5 μM particle size) coupled with a corresponding guard column (8 × 4 mm, 5 μM particle size). A solvent system consisting of acetonitrile:water:acetic acid (70:30:0.1, by vol) was used at a flow rate of 1 mL·min^−1^. For more detailed analysis of the product structures the conjugated dienes formed during the incubation period were prepared by RP-HPLC and further analyzed by normal-phase HPLC (NP-HPLC). For these analyses we used the solvent system n-hexane:2-propanol:acetic acid (100:2:0.1, by volume) and analytes were separated on a Nucleosil 100-5 column (250 × 4.6 mm, 5 µm particle size, Macherey-Nagel, Düren, Germany). For control activity assays a lysate supernatant of *E. coli* cells was used, which were transformed with a control expression plasmid. A similar experiment was carried out using the cellular supernatant of Sf9 infected with a control baculovirus.

### 4.8. LC-MS/MS Analysis of the Reaction Products

After incubation of the different bony fish ALOX15 isoforms with arachidonic acid the oxygenation products were prepared by RP-HPLC and further analyzed by LC-MS/MS as described [83].

### 4.9. Kinetic Studies Using Recombinant Bony Fish ALOX-Isoforms

For kinetic characterisation of the different putative bony fish ALOX15 orthologs we determined the following parameters:

(i) pH-profile of AA oxygenation. For this purpose, equal volumes of 10 mM phosphate buffer were mixed with 10 mM borate buffer and different pH-values were adjusted by the addition of 5 N NaOH or 5 N HCl. Activity assays were carried out at the different pH-values and the amount of AA oxygenation products formed at the different pH-values was determined by RP-HPLC. The highest activity was set to 100% and the relative AA oxygenase activities measured at the other pH-values were calculated.

(ii) Temperature profile of AA oxygenation and activation energy. AA oxygenase activity assays were performed at different temperatures and the amount of reaction products formed during the incubation period were quantified by RP-HPLC. Here, again, the highest activity was set 100% and the relative AA oxygenase activities at the other temperatures were calculated. The activation energy of AA oxygenation was determiend using the Arrhenius-equation in the temperature range between 5–20 °C.

(iii) Oxygen dependence: To judge the oxygen affinity of the putative bony fish ALOX15 orthologs we carried out AA oxygenase activity assays under normoxic and hyperoxic conditions. For normoxic measurements, the reaction buffer of the activity assay was saturated with air (approximately 160 µM oxygen). For hyperoxic measurements, the reaction buffer was bubbled with pure oxygen gas for about 30 min (approximately 800 µM oxygen) and the reaction was started by the addition of the enzyme preparation (bacterial lysate supernatants) to the incubation mixture.

### 4.10. Miscellaneous Methods

SDS-PAGE was performed as described [84]. Briefly, 100 µg denatured protein of the bacterial lysate supernatants were run on a 7.5% polyacrylamide gel. The protein concentrations in the bacterial lysates were qunatified using Bradford Reagent for quantitative protein determination (AppliChem, VWR International GmbH, Darmstadt, Germany) according to the instructions of the vendor. Amino acid sequence alignments were performed using the EMBOSS Needle tool (https://www.ebi.ac.uk/Tools/psa/emboss_needle/).

### 4.11. Statistics Evaluation of the Experimental Raw Data

Statistical calculations and figure design were performed using GraphPad prism version 8.00 for Windows (GraphPad Software, La Jolla, CA, USA) or the SPSS software package.

## Figures and Tables

**Figure 1 ijms-23-16026-f001:**
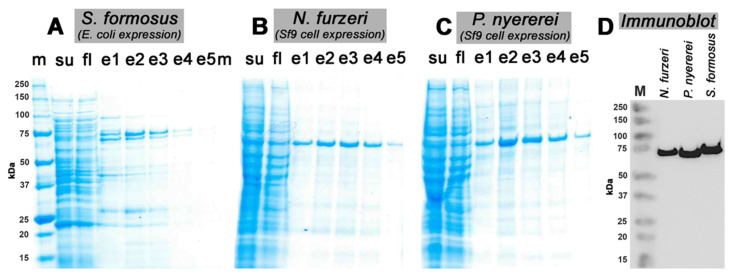
Recombinant expression and purification of bony fish ALOX isoforms. The putative bony fish ALOX15 orthologs coding regions were expressed as recombinant N-terminal his-tag fusion proteins in either *E. coli* (*S. fomosus*) or in Sf9 insect cells (*N. furzeri*, *P. nyererei*) and cellular lysates were prepared. Aliquots of the lysate supernatants were mixed with preconditioned Ni-NTA agarose and the recombinant his-tag fusion proteins were eluted (see Section 4). Elution fractions e1–e5 (0.2 mL each) were collected and aliquots were analyzed by SDS-PAGE. ALOX containing elution fractions were pooled and aliquots were analyzed by Western blotting using an anti-his-tag antibody. (m) Molecular weight markers, (su) Cellular lysate supernatants. (fl) Chromatographic flowthrough. (**A**) SDS-PAGE of the elution fractions of the ALOX isoform of *S. formosus* (*E. coli* expression). (**B**) SDS-PAGE of the elution fractions of the ALOX isoform of *N. furzeri* (Sf9 cell expression). (**C**) SDS-PAGE of the elution fractions of the ALOX isoform of *P. nyererei* (Sf9 cells expression). (**D**) ALOX containing elution fractions were pooled and aliquots of these pools were analyzed by immunoblotting using an anti-his-tag antibody.

**Figure 2 ijms-23-16026-f002:**
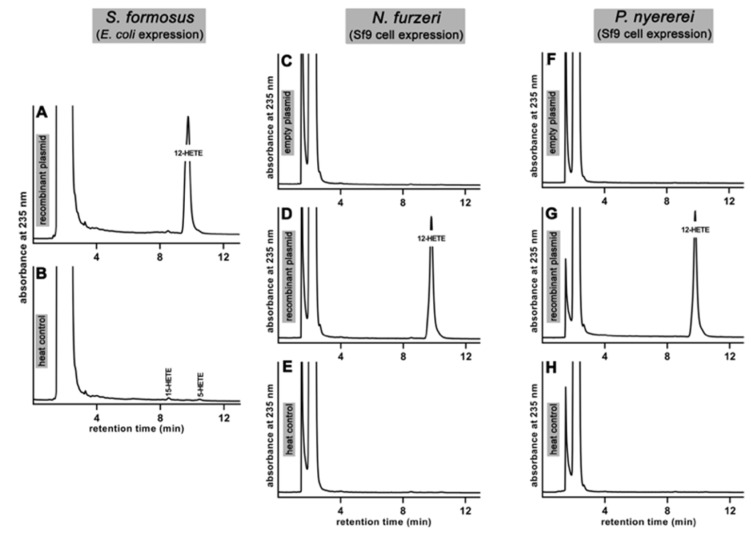
In vitro ALOX activity assays. In vitro ALOX activity assays (see Section 4) were carried out using aliquots of the cellular lysates as enzyme source. The reaction products formed from arachidonic acid during a 3 min incubation period were analyzed by RP-HPLC. Conjugated dienes co-eluting with an authentic standard of 12-HETE are detected (panels (**A**,**D**,**G**)). These products were not formed in two different control incubations [no enzyme controls (addition of PBS instead of cellular lysis supernatant to the activity assay), (panels (**C**,**F**)) and heat controls, (panels (**B**,**E**,**H**))]. Since we normalized each chromatogram to the highest peak direct comparison of the amounts of products formed by the different enzymes is not possible.

**Figure 3 ijms-23-16026-f003:**
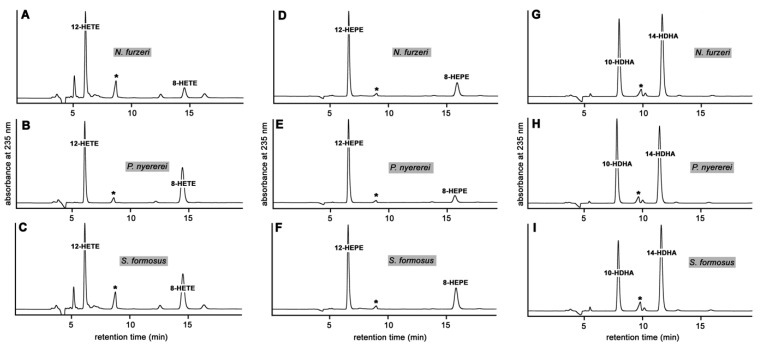
Identification of the major oxygenation products formed from arachidonic acid (AA), eicosapentaenoic acid (EPA) and docosahexaenoic acid (DHA). Activity assays were carried out with AA, EPA and DHA as ALOX substrate (100 µM). The major conjugated dienes formed during the incubation period were prepared by RP-HPLC and further analyzed by NP-HPLC. Product identity was concluded from co-chromatography with authentic standards in NP-HPLC. Two independent oxygenation assays were set up for each enzyme-substrate combination. Lipid extracts were pooled, conjugated dienes were prepared by RP-HPLC and further analyzed by NP-HPLC (see Section 4). (Panels **A**–**C**) AA oxygenation products, (panels **D**–**F**) EPA oxygenation products, (panels **G**–**I**) DHA oxygenation products. * This peak does not show the classical uv-spectrum of a conjugated diene and thus may not be considered a fatty acid oxygenation product. For better comparison of the product profiles each chromatogram was scaled to the highest conjugated diene peak and thus, direct comparison of the amounts of products formed by the different enzymes may not be possible.

**Figure 4 ijms-23-16026-f004:**
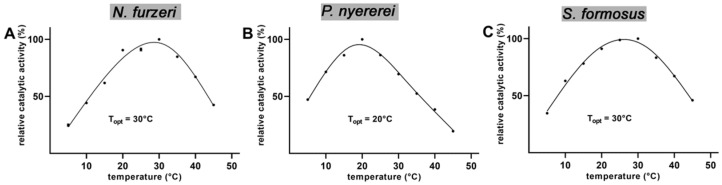
Temperature dependence of ALOX activity of the randomly selected bony fish ALOX isoforms. Activity assays were carried out using the cellular lysate supernatants as enzymes source. Aliquots of the lysate supernatants were incubated in the standard activity assays with AA as substrate at different temperatures and the amounts of specific ALOX products formed during a 3 min incubation period were quantified by RP-HPLC (see Section 4). At each temperature two independent incubations were carried out. Means and error bars are shown, and curves were calculated by non-linear regression (**A**) *N. furzeri*, (**B**) *P. nyererei, (***C**) *S. formosus*.

**Figure 5 ijms-23-16026-f005:**
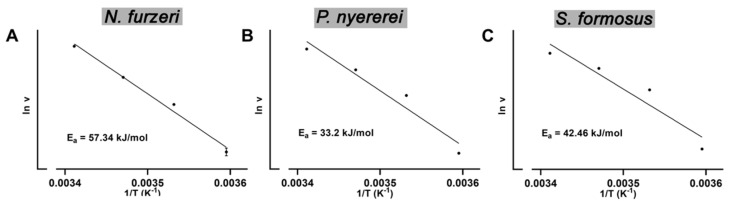
Activation energies for AA oxygenation catalyzed by the selected bony fish ALOX isoforms. The temperature profiles for AA oxygenation by the randomly selected bony fish ALOX isoforms were determined as indicated in Figure 4. The catalytic activities assayed in the temperature range between 5 °C and 20 °C were used to construct the Arrhenius diagrams and from the slope of the resulting lines the activation energies were evaluated for the different bony fish ALOX isoforms. At each temperature two independent activity assays were carried out and thus, the calculated activation energies for each enzyme are based on eight independent experimental data points. (**A**) *N. furzeri*, (**B**) *P nyererei*, (**C**) *S. formosus*.

**Figure 6 ijms-23-16026-f006:**
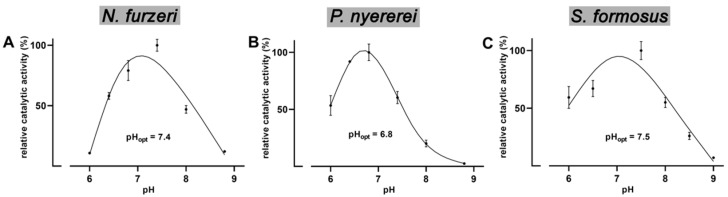
pH-dependence of AA oxygenation catalyzed by the different bony fish ALOX isoforms. The lysate supernatants were used as enzyme source and activity assays were carried out at different pH values. As reaction buffers equimolar mixtures of 10 mM phosphate buffer and 10 mM borate buffer were used and different pH-values were adjusted with 5 N HCl or 5 N NaOH at room temperature. The amounts of ALOX products formed during a 3 min incubation period were quantified by RP-HPLC Two independent activity assays were carried out at room temperature at each pH and means and error bars are indicated. The catalytic activity at pH_opt_ for each enzyme was set 100%. (**A**) *N. furzeri*, (**B**) *P. nyererei*, (**C**) *S. formosus*. Correlation coefficients R^2^ > 0.9 (R^2^ for *N. furzeri* = 0.971, R^2^ for *P. nyererei* = 0.986, R^2^ for *S. formosus* = 0.951).

**Figure 7 ijms-23-16026-f007:**
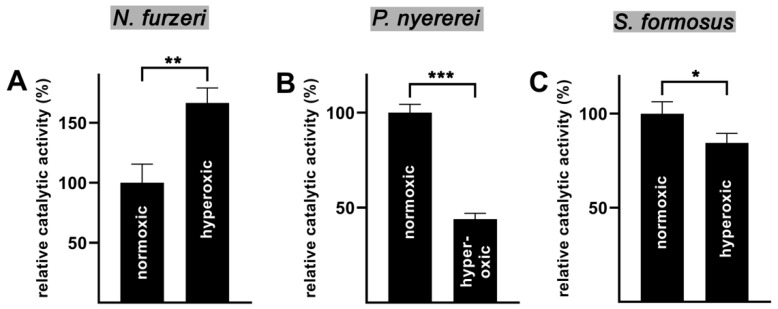
Oxygen dependence of AA oxygenation catalyzed by the three bony fish ALOX isoforms. Activity assays were carried out under normoxic and hyperoxic conditions and the formation of specific ALOX products during the incubation period (3 min) was quantified by RP-HPLC. Three independent measurements were carried out for each enzyme at either condition. The mean of product formation under normoxic conditions for each enzyme was set 100% and the relative oxygenase activities were calculated. Experimental raw data were evaluated using the two-sided *t*-test. (**A**) *N. furzeri,* (**B**) *P. nyererei*, (**C**) *S. formosus*. The asterixis above the columns indicate the degree of statistical significance (* *p* < 0.05, ** *p* < 0.01, *** *p* < 0.001, *t*-test).

**Figure 8 ijms-23-16026-f008:**
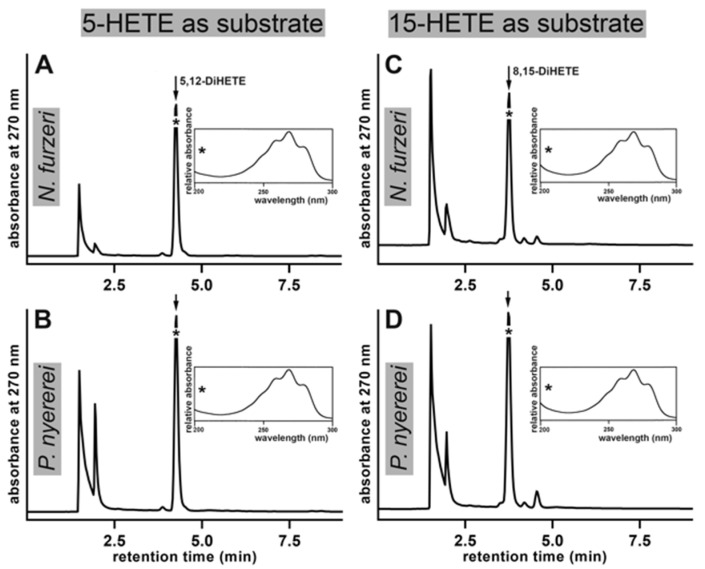
Oxygenation of 5*R/S*-HETE, 15*R/S*-HETE by bony fish ALOX-isoforms. ALOX activity assays (substrate concentration 30 µM) were carried out using the cellular lysate supernatants as enzyme source. After 10 min of incubation the reaction products were reduced and analyzed by RP-HPLC recording the absorbance at 270 nm. The uv-spectra of the major oxygenation products were taken during the chromatographic run at the time points indicated (*). The chemical structures of the major oxygenation products were confirmed by LC-MS analysis. (**A**) *N. furzeri* ALOX, 5-HETE as substrate, (**B**) *P. nyererei* ALOX, 5-HETE as substrate, (**C**) *N. furzeri* ALOX, 15-HETE, as substrate (**D**) *P. nyererei* ALOX, 15-HETE as substrate. For better comparison of the product profiles each chromatogram was scaled to the highest conjugated triene peak. Thus, the amounts of products formed by the two enzymes cannot directly be compared.

**Figure 9 ijms-23-16026-f009:**
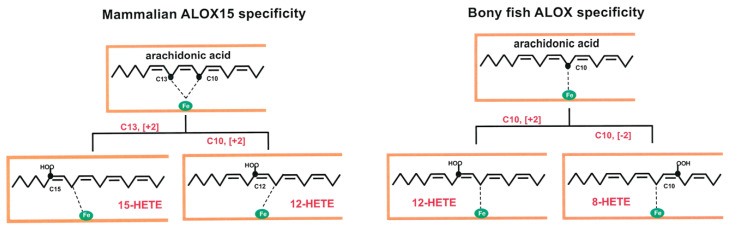
Bony fish ALOX isoforms exhibit a different kind of dual reaction specificity when compared with mammalian ALOX15 orthologs. Mammalian ALOX15 orthologs catalyze initial hydrogen abstraction from two different bisallylic methylenes (C_13_ and C_10_ of AA) and the fatty acid radical undergoes [+2] radical rearrangement. Thus, 12- and 15-HETE are simultaneously formed. Bony fish ALOX-isoforms catalyze hydrogen abstraction from a single bisallylic methylene (C_10_), but the fatty acid radical formed rearranges in opposite directions ([+2] and [−2]).

**Table 1 ijms-23-16026-t001:** Substrate specificity of bony fish ALOX-isoforms with oxygenated polyenoic fatty acids. Different oxygenated metabolites of polyenoic fatty acids were used for comparative activity assays. For these assays the different substrates were incubated in PBS (30 µM substrate concentration) and the rates of substrate conversion during a 10 min incubation period was assayed by RP-HPLC. The rate of AA conversion was set 100% and the relative conversion of the other substrates was calculated. Cellular lysate supernatants were used as enzyme source and identical amounts of enzyme were added to the different substrates. n.i. not identified.

Substrate	Relative Oxygenase Activity (%)			Main Product
	*N. furzeri*	*P. nyererei*	*S. formosus*	
AA	100	100	100	12S-HETE
5(R/S)-HETE	209.9	242.6	393.3	5,12-DiHETE
15(R/S)-HETE	78.2	202.4	508.3	8,15-DiHETE
18(S)-HEPE	9.5	20.9	1.0	n.i.
18(R)-HEPE	3.7	4.8	1.0	n.i.
5(S),15(S)-DiHETE	1.0	1.0	1.0	n.i.
5(S),6(R)-DiHETE	1.0	1.0	1.0	n.i.

## Data Availability

Supportive information for this paper is deposited at Appendix A. Experimental raw data will be provided by the authors to interested readers upon request.

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
