# Peer review of "Functional Characterization of Novel Bony Fish Lipoxygenase Isoforms and Their Possible Involvement in Inflammation"

_ijms, 2022, doi:10.3390/ijms232416026_

Round 1

Reviewer 1 Report

The authors of the above-mentioned manuscript selected three lipoxygenase genes from bony fish in the NCBI protein database for heterologous expression and subsequent enzyme characterization. The authors managed to overexpress the active enzymes as His-tag fusion proteins in either Escherichia coli or the Sf9 cells-baculovirus system. The authors were unable to purify the recombinant enzymes in active form. The enzymes in the crude extracts were characterized in terms of temperature and pH dependence of the activities, substrate specificity, activation energy and KM-values.

Major issue:

Probably, the selected immobilized Ni-binding ligand exhibited high affinity to the lipoxygenase-bound iron, thereby depriving the enzymes of the essential metal ion, causing inactivation of the enzymes. Maybe it is not Fe that is bound in the recombinant lipoxygenases, other metal ions are conceivable.

Line 256: In my opinion, it is quite a cheap statement, when you say that you “were forced to use the cellular lysate” instead of purified enzymes. There are a number of alternatives to the His-tag technology available (Mahmoodi et al. (2019) Current affinity approaches for purification of recombinant proteins, CogentBiology, 5:1; Mishra et al. (2020) Affinity tags for protein purification, Current Protein & Peptide Science, 21, 821–830). Solving the purification problem encountered here for the lipoxygenases would help other researchers in their projects. For a journal with such a relatively high impact factor, one would expect more effort to solve the purification problem. In my opinion, the novelty and the offered amount of data in the current manuscript does not warrant publication in International Journal of Molecular Sciences.

Further issues, remarks:

-         Line 256: The authors report that the purification procedure inactivated the enzymes; please, specify if this holds true for all three enzymes.

-         Line 463: How did you obtain the di-HETE compounds?

-         Figure 7B, line 421: not significant? “As shown in Figure 7 we did not observe significant differences between product formation under normoxic and hyperoxic conditions for any of the three ALOX-isoforms.” The authors do not explain the outcome of the Figure 7B experiment. A half-maximum activity at hyperoxic conditions compared to normoxic conditions is, in my opinion, significant. What caused this decrease in activity? (enzyme denaturation ?)

-         The activation energy data are of quite limited value.

Author Response

see attached document

Reviewer 2 Report

The manuscript “Functional characterization of novel bony fish lipoxygenase isoforms and their possible involvement in inflammation” describes the heterologous production of three different arachidonic acid lipoxygenases from bony fish species as well as their characterization.

The manuscript is well written and mostly easy to understand, but leaves some points open that have to be addressed before publication is possible.

The focus on ALOX15 remains unexplained. Why this isoform and none of the others? The differences between the isoforms also remains somehow dubious. Maybe a table would help?

The first paragraph of 3.1 (ll. 223-226) already mentions the fact that two of the proteins could not be obtained using E. coli, which is then explained in detail in the subsequent paragraph. This should be changed to a chronological explanation to simplify understanding.

The figure legends in general explain a lot of details that belong in the Material and Methods, not into a figure legend.

I understand that it was necessary to use the cell lysate for the experiments as the purification failed. Were no other purifications tried? I also do not understand how you made sure to use equal amounts of the different enzymes without any purification. Is this the reason all the data is given normalized to 100 %? I would have liked to be able to actually compare the activities between the three enzymes. Finally, at least once, controls with the corresponding empty vectors that show no relevant activity should be shown (e.g. in the supplement) to ascertain the reader that all activity documented is due to the ALOX and not any of the other proteins in the lysate.

Figures 2 and 8 are missing the numbers on the y-axis. This way, no comparison between the subgraphs is possible.

Figure 2: What is the peak at 5 minutes in A and C? And the second peak after the *-peak in G to I?

Ll. 313-316: This belongs in the discussion!

In the legend of Fig. 3, it is suddenly “randomly selected bony fish”, which was never mentioned before. Should this not be mentioned at a more prominent position in the introduction?

Ll. 335-310: This also belongs in the discussion.

Figure 4: R2-values as well as the numbers on the y-axis are missing. Are error bars shown? The legend does not say.

Figure 5: The fit for A and C is really bad. Please add the respective R2-values if you really want to fit these curves.

Ll. 371-373 should also be in the discussion.

Ll. 375-380: This is general information that every reader should be aware of.

Figure 6: You cannot fit a saturation curve without having reached the saturation! Thus, you are able to fit C, but A and B have to be tested with higher substrate concentrations for a Km-determination! The fit right now is horrible. In addition, the way you normalize the data to 100 % is highly misleading as 100 % in a Km-graph should be the saturation and clearly is not, if you compare your values at 50 % and the mathematically calculated Km-values. Also, what about the two (A) and three (B) data points at the low concentrations? Right now, these are simply ignored. What happened there?

Ll. 394-397 should be in the discussion. Why did you choose to test your system without detergents if that is normally what is done?

Figure 7: I find it extremely surprising that A and even more so B are not significantly different. Can you give more detail on your significance testing?

From Figure 8 on, the incubation time has suddenly been increased from 3 to 10 min. Can you explain why?

Figure 8: The first peak at 1.5 min in C and D is bigger than the one at 4 min, which had been annotated as 8,15-HETE. What is it and why is it not regarded as major product?

L. 528: This is not true for N. furzeri.

Ll. 532+533: Exactly, this comparison should be elucidated on here.

L. 616: Why were these experiments not performed?

Last but not least: There are an enormous amount of 89 references for a single paper. Please check carefully if all of them are really necessary.

To finish, some minor points regarding formatting and grammar/spelling errors:

·         L. 22: oxygenation – Wrong hyphenation

·         Formatting in 2.2 is wrong

·         L. 193: Was set to 100 %

·         Addition of the respective species the ALOX originates from to the subfigures would help the reader instead of always having to check the figure legend.

·         Figure legend of Fig. 1: The formatting has gone wrong here.

·         L. 273: What is TMS? Please add the full name.

·         Ll. 332+333: It should be . instead of , in English.

·         L. 560: An orthologous gene – you name six!

·         L. 608: However should not be bold.

Author Response

see attached document

Round 2

Reviewer 1 Report

Single major issue regarding the manuscript IJMS 2042103 after revision:

It is still my opinion that the manuscript would fundamentally benefit from a working protein purification protocol for the heterologously produced lipoxygenases. Without proper protein purification, the authors ought to show that the host cells (without recombinant protein production) do not exhibit the examined activities. In my opinion, this is the minimal requirement of a scientifically sound work. If I am not mistaken, such controls are missing in the manuscript.

Author Response

Reviewer 1
Comment of reviewer 1: The authors should show that host cells (without recombinant protein production) do not exhibit the examined activities.
Response of authors: To address the comment of the reviewer we carried out an additional expression experiment testing the ALOX activity of a cellular lysate prepared from E. coli, which were transformed with an empty expression plasmid and from Sf9 infected with a control (non-ALOX) baculovirus. This image was added as supplemental figure to the ms.

Reviewer 2 Report

The manuscript has been thoroughly reviewed and improved. My only problem remains the following: If you are forced to use cell lysates because the purification did not work, you have to prove that the lysate without ALOX has no corresponding activity in your activity assays using a control cell lysate from cells containing the empty vector. In your case, this would be two controls, one for each cell type. A heat-inactivated control or one without any enzyme is not sufficient. Thus, the control is lacking and – in my honest opinion – the manuscript is not publishable without it.

Author Response

Reviewer 2

Comment of reviewer: The authors have to prove that the lysates without ALOX has no corresponding activity in your activity assay using a control cell lysate from cells containing the empty vector.

Response of authors: To address the comment of the reviewer we carried out additional experiments testing the ALOX activity of cellular lysate prepared from E. coli which were transformed with the empty expression plasmid and from Sf9 infected with a control (non-ALOX) baculovirus. This image was added as supplemental figure to the ms.

Round 3

Reviewer 1 Report

I am sorry, I cannot find Figure S1 in the supplementary files. 

Author Response

We submitted the supplemental material as a compressed .zip file. Wehn you decompress this file there should be a the supplemental Table S1 and the Supplemental Figure S1. There may hav been some problems with the compression procedure and thus, we resubmitting the supplemental files again. The ms itself has not been aletered.

Reviewer 2 Report

Dear Sir or Madam,

the mentioned additional figure could not be found. The document called 'supplement' again included the new revised manuscript. Please add a reference to the new supplementary figure in the main text and add the supplementary figure to the supplement.

Author Response

We submiited the supplementary material that involves the supplemental Table S1 and the supplemental Figure S1 as combined .zip file. Unfortunately, there must have been a problem with the compression process so that other files have been included. Thus, we resubmit the correct supplement  again. The ms file itself has not been changed.